# Graphene Oxide (GO) Materials—Applications and Toxicity on Living Organisms and Environment

**DOI:** 10.3390/jfb13020077

**Published:** 2022-06-10

**Authors:** Aminah N. Ghulam, Otávio A. L. dos Santos, Layla Hazeem, Bianca Pizzorno Backx, Mohamed Bououdina, Stefano Bellucci

**Affiliations:** 1Department of Biology, College of Science, University of Bahrain, Zallaq P.O. Box 32038, Bahrain; aminahnasar1995@gmail.com (A.N.G.); lhazeem@uob.edu.bh (L.H.); 2Instituto de Bioquímica Médica, Universidade Federal do Rio de Janeiro, Rio de Janeiro 21941-902, Brazil; otavioldoss@gmail.com; 3Numpex-Bio, Universidade Federal do Rio de Janeiro, Campus Duque de Caxias, Duque de Caxias 25245-390, Brazil; biapizzorno@gmail.com; 4Department of Mathematics and Sciences, Faculty of Humanities and Sciences, Prince Sultan University, Riyadh 11586, Saudi Arabia; mbououdina@psu.edu.sa; 5INFN-Laboratori Nazionali di Frascati, Via E. Fermi 54, 00044 Frascati, Italy

**Keywords:** graphene derivatives, health, environment, cytotoxicity

## Abstract

Graphene-based materials have attracted much attention due to their fascinating properties such as hydrophilicity, high dispersion in aqueous media, robust size, high biocompatibility, and surface functionalization ability due to the presence of functional groups and interactions with biomolecules such as proteins and nucleic acid. Modified methods were developed for safe, direct, inexpensive, and eco-friendly synthesis. However, toxicity to the environment and animal health has been reported, raising concerns about their utilization. This review focuses primarily on the synthesis methods of graphene-based materials already developed and the unique properties that make them so interesting for different applications. Different applications are presented and discussed with particular emphasis on biological fields. Furthermore, antimicrobial potential and the factors that affect this activity are reviewed. Finally, questions related to toxicity to the environment and living organisms are revised by highlighting factors that may interfere with it.

## 1. Introduction

The continuous increment in the human population has caused serious issues worldwide, including pollution and contamination of the air and water, which has led to the depletion of natural resources, climate change, and the emergence of numerous diseases, such as those related to respiratory problems and the lack of potable water. In addition, the indiscriminate use and incorrect disposal of antibiotics leads to another problem, namely the rapid growth of antibiotic-resistant bacteria [1,2,3]. Given this situation, different strategies have been investigated to reduce or solve these problems, primarily the development of carbon nanomaterials such as graphene-based materials that have great versatility and exhibit fascinating and tunable properties and applications [1].

Graphene is a material made of carbon atoms that are bonded together by sp^2^ hybridization and arranged in a hexagon pattern, giving it a honeycomb-like structure [4]. Because of its typical 2-dimensional structure, it achieves fascinating and unique numerous characteristics such as being the strongest material, as well as being the lightest, most conductive and transparent material [5]. It is considered as the simplest form of carbon and the thinnest material produced so far [4]. Graphene is rarely used for biological applications because it is insoluble in an aqueous medium due to its high hydrophobicity [1]. However, graphene oxide (GO), a chemically modified graphene [6], presents a single atomic layer [7] and is, thereby, classified as a two-dimensional material [8]. It is basically an oxidized form of graphene laced with oxygen-based functional groups such as hydroxyl (−OH), alkoxy (C-O-C), carbonyl, carboxylic acid (−COOH), and other oxygen-based functional groups at the sp^2^ carbon basal plane [9], which makes it amphiphilic [1]. It contains a high concentration of oxygen; according to C/O characterization, it is less than 3.0 and closer to 2.0 [6]. Therefore, GO is an important material for biological applications because of its interesting properties, including hydrophilicity, high dispersion in aqueous media, simple synthesis, robust size, high biocompatibility, low cytotoxicity, and surface functionalization due to the presence of functional groups. In addition, GO can interact with biomolecules such as proteins and nucleic acid [1], as well as used in nanocomposite materials [9].

However, despite high colloidal stability in water and a unique set of mechanical, colloidal, and optical properties, it is known that the adopted methods for the synthesis of GO utilize strong oxidants, such as potassium permanganate, leading to significant amounts of defects in its crystalline network [10]. These affect GO’s conductive properties, becoming far lower than those of graphene, although its optical and mechanical properties suffer a lesser impact [10]. To resolve this issue, it is possible to perform treatments capable of acting on GO, such as producing reduced graphene oxide (rGO) and, consequently, restoring graphene-like properties [10]. The major motivation for the synthesis of rGO is because of its facile fabrication and processing, as well as because it possesses many improved properties and the ability to be incorporated into many applications [4].

Therefore, GO can be treated using different methods, such as thermal, chemical, and photo-irradiation, to minimize its oxygen content to produce rGO. Herein, it is important to highlight that graphene derivatives such as GO and rGO (Figure 1), due to their ideal material properties and dispersibility in polymer matrices, can be effectively used in polymer nanocomposite materials, which results in an additional broad range of applications [9].

Graphene-based nanomaterials are receiving great attention due to their potential applications in many fields, such as biomedicine, biotechnology, and environmental technologies. However, many studies have reported that they could manifest toxicity to biological systems, which can be influenced by different factors such as lateral size, surface structure, functional groups, purity, dosage, and exposure time. Graphene-based nanomaterials could cause in vivo and in vitro toxicity in animals, plants, and microorganisms, associated with their ability to invade through cellular structures or barriers by several exposure approaches and entry pathways to the body or cells. The toxicity depends on many factors such as different exposure ways and entry pathways, various tissue distribution and excretion, and different cell uptake patterns and locations [11]. Given this scenario and the variety of applications of GO and rGO, it is necessary to expand the knowledge of their toxicity, as well as contamination routes and possible interactions, to avoid damage to the environment in addition to human and animal health in particular, yielding difficulties for many applications such as agriculture and medicine [12]. In this review, the synthesis methods of graphene derivatives and how they can interfere with their properties are presented and discussed. In addition, while exploring their potential applications, it is important to highlight the limitations associated with the serious concerns related to their toxicity.

## 2. Evolution of GO Synthesis Methods

GO was first synthesized in 1859 by Brodie by adding potassium chlorate to a mixture of graphite in fuming nitric acid. In 1898, Brodie’s mechanism was improved by Staudenmaier to produce a simple and revised procedure to fabricate highly oxidized GO by using concentrated sulfuric acid and fuming nitric acid and adding chlorate to the mixture. Lately, in 1958, Hummer’s method was developed to prepare GO that could be used for producing large graphitic films. The most popular Hummer’s method uses KMnO_4_ and NaNO_3_ in concentrated H_2_SO_4_, compared to previous methods. It requires less time for the reaction to occur with high reaction efficiency. Additionally, Hummer’s method provides reaction safety by using KMnO_4_ instead of KClO_3_ to avoid producing harmful byproducts, and it uses NaNO_3_ instead of fuming HNO_3_ to eliminate the formation of acid fog [13]. Nevertheless, Hummer’s method still has several drawbacks, such as the release of toxic gases (NO_2_ and N_2_O_4_), low yield [14], and the generation of Na^+^ and NO_3_^−^ ions, which are harder to eliminate from the wastewater formed from the procedure of synthesizing and purifying GO [13].

Currently, various strategies have been developed to improve Hummer’s method to address these drawbacks: (i) the removal of NaNO_3_; (ii) the addition of a peroxidation step before KMnO_4_ oxidation in the absence of NaNO_3_ (in this regard, Kovtyukhova et al. reported that the peroxidation of graphite with K_2_S_2_O_8_ and P_2_O_5_ before oxidation resulted in the production of highly oxidized GO, but the whole process was time-consuming); (iii) the increment of the amount of KMnO_4_ instead of NaNO_3_; and (iv) the removal of NaNO_3_ and the replacement of KMnO_4_ with K_2_FeO_4_ [14]. Marcano et al. determined that the removal of NaNO_3_ increments the amount of KMnO_4_, whereas conducting the reaction in a 9:1 mixture of H_2_SO_4_:H_3_PO_4_ improved the efficiency of the oxidation process and provided a large amount of hydrophilic oxidized graphene material (oxidized GO). In addition, this modification avoided the generation of toxic gases, since the temperature was controlled easily to produce a large amount of GO [15].

Even with the implemented modifications, there are still some flaws with Hummer’s method: (i) the high utilization of the oxidants and intercalating agents; and (ii) the time-consuming process of synthesis, which leads to high cost and poor scalability in practical applications. Recently, additional modifications were introduced to the existing NaNO_3_-free Hummer’s method: (i) the replacement of KMnO_4_ with K_2_FeO_4_ of higher oxidability at a low temperature to improve the intercalation and preoxidation of graphite; (ii) the two-step feeding of KMnO_4_ to raise the consumption of the oxidants; and (iii) an increase in the concentrations of the graphite and oxidants by lowering the amount of concentrated H_2_SO_4_ to synthesize GO in an economical, eco-friendly, and large-scale approach [14].

In the past few years, various techniques have been developed for the synthesis of graphene and graphene oxide materials, but many of these are highly sophisticated and expensive. For this reason, most of the commercially available GO samples are synthesized using Hummer’s method or its modified version [16]. More recently, graphene oxide was produced by directly oxidizing sugarcane bagasse under a muffled atmosphere. After the juice’s extraction, the fiber was crushed and well-ground to produce a fine powder. This powder was mixed with ferrocene and placed directly into a muffle furnace at 300 °C for 10 min under atmospheric conditions to produce a black solid. The performed analyses showed that the sugarcane bagasse was fully oxidized into graphene oxide [17].

## 3. Reduction of GO

GO reduction is a simple and inexpensive way to produce materials with graphene-like characteristics [10]. It has been reported that various GO reduction methods have resulted in different properties of rGO, which have a direct influence on the final performance of materials or devices composed of rGO [18]. For example, since a reduction process can improve the electrical conductivity of GO, increased charge carrier concentration and mobility improves the reflection of the incident light, which makes rGO films exhibit a metallic luster compared to GO film precursors with brown color and semitransparent character (Figure 2A) [18]. Reduction in a colloid state by chemical reduction usually results in a black precipitation from the original yellow-brown suspension (Figure 2B), which is probably a result of enhancement in the hydrophobicity of the material caused by a decrease in polar functionality on the surface of the sheets [18]. Additionally, GO’s chemical composition usually ranges from C_8_O_2_H_3_ to C_8_O_4_H_5_, depending on the preparation method, with a C/O ratio of 4:1–2:1. After reduction, the C/O ratio can be improved to approximately 12:1 [18].

GO contains oxygen, mainly in the form of epoxy (bridge site oxygen), hydroxyl (−OH groups), and carboxylic acid (−COOH) groups. The value of the C:O:H ratio in GO is dependent on the adopted synthetic paths, the degree of oxidation, and different synthesis conditions. On average, the percentage of oxygen in GO remains around 30% by weight. There are many methods for the reduction of GO, including (1) chemical reduction, (2) thermal reduction, and (3) solvothermal reduction.

### 3.1. Chemical Reduction

It is possible to reduce GO using different reducing agents, including hydrazine hydrate, dimethylhydrazine, sodium borohydrate (NaBH_4_), hydrogen plasma, and urea. Additionally, other biomolecules can be utilized through an eco-friendly process, such as amino acids, carbohydrates, vitamins, and plant extracts [21,22]. The most common method is the treatment of GO with a hydrazine hydrate solution at 100 °C for 24 h, but to shorten the duration time, GO can be exposed to hydrazine vapor. The usage of NaBH_4_ hydrolyzed in water at a slow rate is considered to be a more effective method for reducing C=O groups in GO, but it exhibits low-to-moderate efficiency for reducing epoxy, carboxylic acids, and alcohol groups. The fastest reduction method is the exposure of GO to hydrogen plasma. However, an inexpensive and simple reduction method is heating GO with urea [23].

### 3.2. Thermal Reduction

Thermal reduction may occur by the direct heating or irradiation (microwave, ultraviolet, or infrared visible) of GO under a vacuum, inert, or reducing atmosphere. Thermal reduction is a good method for simultaneously eliminating oxygen-based groups and repairing the structure of GO by thermal annealing. It helps for the reparation of oxidation defects of the GO carbon basal plane. The reduction of GO can occur at a broad range of temperatures. The duration of the time required for reduction to occur depends on the temperature, so at 50 °C, several days are required for the reduction to occur. At temperatures above 2400 °C, which can be achieved with direct Joule heating, reduction occurs in less than one minute. In general, reduction occurs at temperatures in the range of 400–1200 °C [24].

The binding energy between graphene and different oxygen-containing functional groups can be a relevant factor to evaluate the reducibility of each group attached to the carbon plane, especially during the thermal deoxygenation processes. Kim et al. calculated using density functional theory (DFT) the binding energies of an epoxy group (62 kcal/mol) and a hydroxyl group (15.4 kcal/mol) to a 32-carbon-atom graphene unit. These obtained values indicated that epoxy groups are much more stable than hydroxyl groups in GO [25]. Similarly, the calculations performed by Gao et al. [26] reported that the epoxy and hydroxyl groups in GO could be divided into two types based on their different locations at either the interior of an aromatic domain of GO or the edge of an aromatic domain. A single hydroxyl group attached to the interior aromatic domain has low binding energy, and for this reason, it is not stable, and dissociation may occur at room temperature, whereas a hydroxyl group attached to the edge is stable under the same conditions. Additionally, it is estimated that the critical dissociation temperature of hydroxyl groups attached to the edges of GO is 650 °C and that only above this temperature can hydroxyl groups be fully removed. For carboxyl groups, it is expected that they are reduced at 100–150 °C, while carbonyl groups are more stable and only reduce for temperatures above 1730 °C [18].

### 3.3. Solvothermal and Hydrothermal Reduction

Solvothermal and hydrothermal reduction represents a combination of chemical and thermal methods under supercritical conditions or nearby pressure–temperature domains resulting from heating [27]. This reduction occurs within a sealed container by enhancing the surface reactivity under high pressure and moderate temperatures [27]. One method developed uses N-methyl-2-pyrrolidinone (NMP) as a solvent, which has a high boiling point. The reduction of GO occurs due to moderate thermal annealing and the oxygen-scavenging properties of NMP at high temperatures [28]. Compared to other solvothermal reduction methods, this route possesses a lower C/O ratio of reduced GO, as well as the ablity to produce a moderate amount of rGO [18]. In an alternative method, supercritical water is used as a reducing agent in hydrothermal conditions and offers a green chemistry solution to organic solvents. Supercritical water removes the functional groups attached to GO and recovers the aromatic structures of carbon lattice [29]. Green synthesis is a promising approach because it is eco-friendly, inexpensive, simple, and fast; in addition, it reduces toxicity, energy demand, and by-product formation. Moreover, no toxic solvents are used in this process [30].

## 4. Factors Affecting the Properties of GO

The nature and number of functional groups on GO sheets can determine its characteristics, such as band gap energy, transparency, optical and electrical properties, and surface charge. The size of GO sheets is influenced by the functional groups and defect sites, which can be increased by a high degree of oxidation, resulting in the breakdown of GO sheets throughout the exfoliation process. Additionally, the oxidation degree of graphite influences the size of GO sheets. It has been revealed that, by changing the degree of oxidation of graphite, the duration of oxidation, and the quantity of oxidants, GO sheets can be produced with varying amounts of oxygen, which can result in giving variety in the size, electrical conductivity, and energy band gap of GO sheets. Kang and Shin [31] conducted a study to investigate the relationship between the oxidation temperature and the sizes or properties of GO sheets. Different GO sheets were produced using a modified Hummer’s method to evaluate the oxidation temperature’s effect on their surface charges, sizes, C/O ratios, and optical properties. The authors noticed different C/O ratios at different temperatures of 1.18 at 35 °C, 1.24 at 27 °C, and 1.26 at 20 °C, which showed that, at higher temperatures, more functional groups containing oxygen formed on the GO sheets, which increased the amount of oxygen while decreasing the C/O ratio. After the exfoliation, the GO sheets’ lateral sizes were 12.4 μm at 20 °C, 10.8 μm at 27 °C, and 8.3 μm at 35 °C. This determined that, as the oxidation temperature increased, the C/O ratio and the average size of the GO sheets decreased. Moreover, the authors reported that the surface charges and optical properties of the GO sheets were related to the degree of oxidation. The addition of functional groups, especially hydroxyl and carboxylic which are situated at the edge side of the basal plane of the GO sheets during the oxidation process, could weakly generate negative charges in the solution due to deprotonation, giving a hydrophilic nature. Furthermore, as the oxidation temperature increased, the zeta potential of the GO sheets increased, too. Therefore, it was determined that, as the oxidation temperature increased, more functional groups with negative charges were introduced into the GO sheets. Finally, these results demonstrated that size and physical properties, especially the surface charge of the GO sheets, could be controlled through the oxidation temperature by applying a modified Hummer’s method. In addition, this study determined that the properties of GO and rGO could be tuned, which is an important quality for actual applications [31]. Concerning the optical absorption properties of GO, Song et al. reported that GO had good absorption in the visible range of (380–800 nm), but in the ultraviolet range, the absorption slightly decreased. This determined that GO had good photo-response in the visible and ultraviolet ranges, which indicates the great potential of GO for light applications [32].

## 5. Applications of GO and rGO

### 5.1. GO as Antimicrobial Agent

Infections caused by bacteria can result in acute or chronic illness, and the decontrolled use of antibiotics leads to the emergence of bacteria more resistant to them, which manifests a serious health problem around the world [33]. Graphene-based materials (including graphite, graphite oxide, GO, and rGO) possess a wide range of antibacterial activities toward bacteria, fungi, and viruses [34]. Specifically, GO has been reported as a promising material that can be used as an antimicrobial agent and, hence, applied in applications where it is necessary to inhibit microbial growth, such as in the treatment of infections, the coating of medical devices, packaging, and fabrics, among others [35]. Chang et al. reported that GO did not have a significant cytotoxicity effect on lung adenocarcinoma human cells (A549). However, it could produce dose-dependent oxidative stress in cells and a slight loss in cell viability at high concentrations. The cytotoxic effects were concentration- and size-dependent for GO, which is important to consider during the fabrication of antimicrobial GO therapeutics and bio-applications [36].

The growth rate of bacterial cells in the presence of GO depends on the nature of GO, the incubation time, and the dosage of GO nanomaterial. *Klebsiella pneumonia* is a Gram-negative bacterium that attacks mammalian lung cells, damages the tissues, and causes the inflammation of organs, which increases the death rate. However, the in vivo and in vitro studies conducted by Zhao et al. reported that GO inhibited the growth of this bacteria, thereby increasing the survival rate [1]. In 2010, the antibacterial properties of graphene-based nanomaterials were first demonstrated. Graphene-based material (GO and rGO) could inhibit the growth of *Escherichia Coli* [33]. Fallatah et al. investigated the effect of GO nanoparticles on *Pseudomonas putida* KT2440 biofilms of different maturity stages (24, 48, and 72 h). The obtained results demonstrated that GO reduced the viability of 48 h biofilm and detached biofilm cells, which relates to membrane damage, but the viability of the 24 and 72 h biofilms was not affected, and there were no detached biofilm cells. Therefore, this showed that the effect of the antibacterial activity of GO on the viability of biofilms or planktonic cells depended on its stage of maturity, which is possibly due to changes in the physiological state of the cell at different maturation stages. However, further investigation is required for a better understanding of these results [37]. Additionally, studies have demonstrated that GO nanomaterial is a successful agent against multidrug-resistant pathogens, but to produce efficient antimicrobial GO therapeutics, an in-depth investigation must be carried out to better elucidate the antimicrobial mechanism of GO and to highlight the factors affecting the antimicrobial nature of GO [1]. Overall, antimicrobial activity has been reported against different microorganisms, such as Gram-positive bacteria, Gram-negative bacteria, and fungi, and has demonstrated a broad spectrum of actions (Table 1).

#### 5.1.1. Antimicrobial Mechanism of GO

Antimicrobial effects cause a loss in cell viability, oxidative stress, and membrane stress that results in DNA fragmentation involving three steps: (i) cell deposition on graphene materials; (ii) membrane stress due to direct contact with sharp GO nanosheets; and (iii) the production of superoxide radical anions when the cell is exposed to GO and rGO (Figure 3) [49]. Liu et al. compared the antibacterial activity of graphite, graphite oxide, GO, and rGO on *E. coli* under the same concentration and incubation duration. The obtained results demonstrated that GO had the highest antibacterial activity because it caused the highest oxidative stress level (had the most functional groups containing oxygen), followed by rGO, graphite, and graphite oxide [41]. The antimicrobial activity of GO involves physical and chemical factors. The physical factor (Figure 4) involves damage of the cell membrane by the direct contact of the sharp edges of GO nanosheets with the membrane, piercing through it, which causes leakage of the intracellular matrix eventually causing the death of the bacterial cell (cutting effect) [50,51]. The connection between nanosheets and cells occurs through three mechanisms: (i) the swing mechanism, where nanosheets collide with the cell membrane multiple time; (ii) the nanosheets trap the cell membrane by Van der Waals forces and hydrophobic interaction, causing membrane damage; and (iii) the extraction mechanism, which results in distortion of the membrane and the loss of its integrity [1].

Moreover, GO can establish antimicrobial activity for pathogenic bacteria by the GO’s wrapping mechanism (Figure 4). During this process, GO traps the bacterial cell by wrapping around it and stopping it from proliferation by disconnecting the cell from its environment, which results in cell distortion and affects cell metabolism [7,49]. In some cases, the GO sheets can be removed by sonication, and bacteria can proliferate and become activated again [49]. This effect is size-dependent: large GO sheets can completely wrap microorganisms, whereas smaller GO sheets leave the cell partially uncovered, which allows the uptake of nutrients and survival [51]. In the trapping effect, microorganisms face aggregated GO materials and are trapped by them, consequently inhibiting their growth [50].

On the other hand, the chemical factor involves the excess production of reactive oxygen species (ROS) that causes oxidative stress, leading to lipid, protein, and DNA damage and, thereby, causing cell death [50]. A study conducted by Chong et al. reported that the exposure of GO to simulated sunlight increased the antibacterial activity. The measurements of ROS indicated that only singlet oxygen (^1^O_2_) was generated by GO’s exposure to simulated sunlight, which contributed to some extent to oxidative stress, causing antibacterial activity. However, the key cause of antibacterial activity was light-induced electron-hole pairs produced on the GO surface, which encouraged the reduction of GO, introducing additional carbon-centered free radicals that also increased the antibacterial activity of GO. Therefore, it can be concluded that oxidative stress caused by GO is mainly ROS-independent, and simulated sunlight speeds up the electron transfer from the bacterial membrane to GO, causing ROS-independent oxidative stress [52].

#### 5.1.2. Factors Affecting the Antimicrobial Nature of GO

The antimicrobial mechanisms of GO initiated from the interaction between GO and bacterial cells, such as physical piercing of the cell membrane, mechanical wrapping of the cell, and ROS generation, are usually influenced by some physical and chemical properties of GO, including lateral size, morphology, degree of oxidation, basal plane, purity, aggregation, and composites [1].

##### Lateral Size

The size of GO nanosheets influences properties such as adsorption, dispersion, and the sharp edges of GO. These are important for GO’s interactions with microorganisms. Lateral size is a key factor in determining the effectiveness of the antimicrobial action of GO [34]. Perreault et al. conducted a study to investigate the effect of GO sheets’ size areas (ranging from 0.01 to 0.65 μm^2^) on antimicrobial activity by using *E. coli* as a model microorganism. The results indicated that, as the size area of the GO sheets decreased, the antimicrobial activity of GO increased four-fold because smaller-sized GO sheets exhibited greater defects [53]. The defects existing on GO sheets allowed more oxygen absorption and higher defect density in smaller GO sheets, which explained their higher oxidative potential [52]. However, larger GO sheets demonstrated good antimicrobial activity when GO interacted with bacterial cells using a mechanical wrapping mechanism. In this case, larger GO sheets trapped or covered the cell fully and more easily to stop the cell from proliferation, which resulted in loss of cell viability [54]. However, the cell inactivation by the wrapping mechanism was reversible upon removing the GO sheets with sonication [52].

In another report demonstrated the relationship between GO size and its antibacterial activity against the Gram-positive bacteria *Streptococcus mutans*. Increasing the size reduced the cutting effect but enhanced the cell entrapment effect, and vice versa. The smallest GO (1295 nm) caused a significant cutting effect and cell breakage, but the cell entrapment effect could hardly be found. However, larger GO sheets (2015 nm) had a stronger cell entrapment effect, but the cutting effect was weaker. The largest GO sheets (4544 nm) had almost no cutting effect and only showed a strong cell entrapment effect. GO with a smaller size had a higher edge density and, thus, had a stronger cutting effect; in contrast, the GO with a larger size had a wider lateral dimension and, thus, had a higher potential to entrap bacterial cells [48]. In another study, Liu et al. also evaluated the role of lateral size on antimicrobial activity in *E. coli* bacteria. They observed a size-dependent trend in which larger GO sheets showed stronger antibacterial activity than smaller ones. The large GO sheets covered cells more efficiently, hence, the cells could not proliferate, resulting in cell viability loss [55].

##### Morphology

It was reported that a wrinkled surface is a typical feature of GO films that possesses great antibacterial properties due to its corrugated nature [56]. It was also observed that wrinkled GO surfaces could interact with the cell membrane more efficiently than planer GO surfaces to reduce cell viability [1]. In a recent study [56], many types of wrinkled-surface GOs with various levels of roughness were prepared to observe their effects on various bacterial species, such as *E. coli, Mycobacterium smegmatis,* and *S. aureus*. The results showed that the antibacterial action of GO was influenced by the ratio of GO surface roughness to bacterial size. The GO with ~500 nm surface roughness exhibited the most effective antibacterial activity against *E. coli* and *S. aureus* due to the matching of the wrinkle size to the bacterial size, whereas GO film with a much higher surface roughness of ~845 nm exhibited the greatest antibacterial activity against *M. smegmatis*. Herein, it is important to highlight that the production of wrinkled-surface GO sheets is a simple and cheap process. Additionally, a wrinkled-surface GO sheet has a more effective ability to entrap the bacterial cell, resulting in tight interaction between them, which leads to membrane stress, causing disruption and intracellular leakage and leading to cell lysis [56].

Zou et al. observed that smooth top-side GO film possessed effective antibacterial activity toward round-shaped *S. aureus* and *P. aeruginosa*, whereas rough bottom-side GO film possessed effective antibacterial activity toward only rod-shaped *P. aeruginosa*. Therefore, both the shape and morphology of bacteria also can contribute to the effectiveness of GO’s antibacterial activity [34].

##### Aggregation

Abiotic external factors may affect antimicrobial activity in two ways: (i) by influencing the aggregation process and bioavailability of GO and (ii) by modifying microorganisms’ behavior. The most-recognized abiotic factor is ionic strength. The reduction in antimicrobial activity that is caused by hindered interaction between GO and microorganisms is due to cations that induce the aggregation of GO [57]. It was reported that GO’s interactions with bacteria were influenced by GO’s concentration, dispersibility charge, and aggregative state. When tested in different solutions, GO could easily aggregate in electrolyte-containing solutions and demonstrated efficient antibacterial action in all solutions at concentrations below 6 μg/mL. In water, as the GO concentration was increased, the GO antibacterial activity also increased. However, in other solutions (NaCl, CaCl_2_, MgCl_2_, and phosphate-buffered saline (PBS)), GO exhibited no influence on microbial growth due to aggregates that covered the GO edges. Meanwhile, for GO at concentrations above 100 μg/mL in NaCl, CaCl_2_, and MgCl_2_ solutions, large-sized aggregates were formed that mechanically trapped the bacterial cell and, thereby, inhibited their growth. In a PBS solution, the GO aggregates promoted bacterial proliferation. It was concluded that GO’s antimicrobial impact could be adjusted by changing the surrounding solutions [58].

##### Basal Plane

The availability of the GO basal plane facilitates interactions with bacterial cells and has a direct influence on the antibacterial activity of GO [1]. Hui et al. conducted a study to investigate the antibacterial properties of GO in Luria-Bertani (LB) nutrient broth. The results indicated that there was an increase in bacterial cell proliferation due to the inactivation of the antibacterial properties of GO. The reason behind inactivation was the noncovalent adsorption of LB components onto GO basal planes [59]. In LB medium, the GO entraps bacterial cells, but once the availability of the GO basal plane is saturated by the growing bacteria, the antibacterial ability is deactivated. However, this deactivation is temporary and can be activated again by increasing the concentration of GO [58]. In another study, a multilayered GO film made by the immobilization of GO sheets on polyethylene terephthalate (PET) showed antibacterial activity as the number of layers increased, where GO positioned itself flat on the substrate, providing more basal plane than GO sharp edges. It can be concluded that the antimicrobial properties of GO depend on its basal plane, where various modes of bacterial deactivation may occur [50].

##### Purity

Barbolina et al. observed that highly purified and diligently washed GO exhibited neutral behavior, which did not motivate nor suppress bacterial growth, whereas poorly purified GO possessed antibacterial properties due to the presence of soluble acidic impurities that could be eliminated by further purification through neutralization with alkaline substrates. Therefore, the purification status of GO is important when dealing with biological systems because the effect of the material can be concealed by the influence of contaminants [60].

##### Composites

Carpio et al. reported that a composition of poly(vinyl-N-carbazole) and GO (PVK-GO) possessed higher antimicrobial activity against planktonic microbial cells, *E. coli, Cupriavidus metallidurans, B. subtilis*, and *Rhodococcus opacus* biofilms than GO alone [61]. GO–metal composites possessed enhanced antibacterial natures and distinctive molecular affinities or selectivities [1]. Whitehead et al. conducted a study to detect the effect of GO–metal hybrids (AgGO and ZnGO) against four types of bacteria (*E. coli, S. aureus, Enterococcus faecium,* and *K. pneumonia*) and showed that AgGO had the most successful antimicrobial activity since the insertion of Ag into GO increased the antimicrobial activity of GO. GO–metal hybrids could be used as advanced antimicrobial agents [62]. Li et al. observed that carbon nano-scrolls composed of graphene oxide-silver nanocomposites possessed ideal, broad antifungal activity against *Candida albicans* and *Candida tropical* [63]. Moreover, Ag nanoparticles (AgNPs) coated with aminoglycoside antibiotic tobramycin were associated to GO. This composite showed antibacterial mechanisms toward multidrug-resistant *E. coli* [64]. GO contributed to cell wall disruption, while AgNPs led to intracellular oxidative stress, and tobramycin obstructed protein production in the bacteria [64]. GO-chitosan nanocomposite films were produced by crosslinking GO with chitosan at a high temperature (120 °C). The antibacterial properties of the nanocomposite films were investigated, and the obtained results showed that these films possessed bacterial inactivation behavior against *E. coli* (Gram-negative) and *B. subtillis* (Gram-positive). The effectiveness of the antibacterial action of these films made them suitable for application in food packaging [65]. GO coatings on metal films such as Zn, Ni, Sn, and steel could improve electrical conductivity. This happens because GO coated on a metal substrate pumps the electrons to fasten the electron transfer, which improves oxygen-containing functional groups on the GO surfaces to form ROS, causing oxidative stress and resulting in efficient antibacterial activity [66]. Stabilizing agents and polymeric coagents can improve the antimicrobial action of GO. In a recent work, pluronic, a bioinert copolymer, improved the dispersion stability of GO and applied osmotic pressure on the cell membrane, consequently enhancing the antimicrobial activity on hypoosmotically challenged cells [57].

The above-stated factors, which are dependent on the type of graphite, indicate that the antimicrobial impacts of GO are affected by its concentration, the incubation conditions, the exposure duration, and the microorganisms’ characteristics [59]. The microorganisms’ size affects the antimicrobial activity of GO. It has been reported that the size of a bacterial cell determines the required GO dosage for the antibacterial activity to occur efficiently. For example, *S. aureus* is smaller (1 μm) than *E. coli* (3 μm), so *S. aureus* requires lower concentrations of GO to cover (trap) the cell, leading to death, when compared to *E. coli* [67]. GO possesses stronger antibacterial activity against *S. aureus* compared to *E. coli* because it has no outer membrane, and it has a thicker peptidoglycan layer that interacts with GO, while *E. coli* has an outer cell membrane and a thinner peptidoglycan layer, so it resists GO antibacterial activity [49,68].

### 5.2. Water Treatment (Purification)

The incorrect disposal of many human activities causes elevated contamination levels, affecting the environment, plants, and animals. Even at lower concentrations, some pollutants are aggressive and can accumulate through the food chain. For this reason, it is essential to treat contaminated waters before disposal into the environment [69]. GO has attracted much attention as a nanosized adsorbent because it can interact with different pollutants in molecular or ionic forms through electrostatic interaction, π–π interaction, and hydrophobic interaction mechanisms. GO directly interacts and then adsorbs at an excellent level towards ions or molecules [70]. Adsorption is a mass transfer mechanism that transfers a substance in a liquid phase to the surface of a solid and connects it to the surface by physical or chemical interaction [71]. GO nanosheets can be used as nanosorbent material; they possess an excellent adsorption property to efficiently eliminate heavy metals such as Pb, Ni, Cr, Zn, Cd, and Cu that originate from pharmaceutical effluents. Adsorption is an efficient and inexpensive method for eliminating heavy metal ions and organic impurities from wastewaters. GO can fully eliminate Cr and Pb ions, but Ni ions can be eliminated gradually by increasing the GO concentration. It was observed that all heavy metals ions were eliminated successfully at a GO concentration of 70 mg at pH 8 [69]. It was reported that poly(amidoamine)-modified GO had the adsorption ability to eliminate Fe(III), Cr(III), Zn(II), Pb(II), and Cu(II) from water or aqueous solutions at room temperature [72]. A GO–ZrO(OH)_2_ composite eliminated As(III) and As(V) from water or aqueous solutions. An rGO–Fe(0)/Fe_3_O_4_ composite removed As(III), Cr(VI), Hg(II), Pb(II), and Cd(II) from water at pH 7.00 and a temperature of 298 K [73]. Rare earth metal ions are poisonous and present in wastewater, and GO was utilized for the adsorption of Gd(III) La(III), Y(III), and Nd(III) for elimination from water [74].

Kovton et al. reported that GO was a promising nanoporous adsorbent or filter for water purification. The combination of GO sheets with commercial polysulfone (PSU) membranes was utilized to effectively eliminate organic impurities from tap water. The results showed that PSU-GO composites demonstrated a more efficient purification ability than benchmark commercial PSU membranes in the elimination of impurities such as rhodamine B and ofloxacin. The adsorption process of impurities on PSU-GO composites obeyed the Langmuir and Brunauer–Emmett–Teller models with unfinished swelling and the intercalation of molecules between GO layers. This approach was simple, and just the least amount of GO was needed, which was directly deposited on the surface of the polymer and then stabilized using microwaves or heat. PSU-GO composites had great stability, demonstrated after 100 h of tests in commercial water cartridges [75].

The main cause of dye pollutants is the textile industry. The dyes and pigments are harmful pollutants of water and are toxic for aquatic species [74]. GO-TiO_2_ composite films were used as filtration membranes to eliminate dye molecules such as methyl orange and rhodamine B from water [76]. Additionally, GO could remove methylene blue, methyl violet, acridine orange, and methyl green dyes from water. The use of graphene-based adsorbents for efficient dye removal was found to be directly proportional to the ionic strength and pH. Due to good electrostatic interactions between GO and dye molecules, the removal process was efficient [74]. Anionic dyes such as AO8 and DR23 were found to be easily adsorbed by GO since both are nonplanar molecules that attach to the skeleton of GO through the formation of π–π stacking interactions due to spatial restriction, whereas electrostatic attraction was the main contribution to the mechanism. This was because the adsorption followed more of a Langmuir model [70]. More interestingly, chitosan was used with GO to synthesize adsorbent hydrogel materials, which have affinities for both cationic and anionic dyes and for heavy metals [77]. Due to the antimicrobial properties of GO, it decreased the GO membrane biofouling while increasing its lifetime and enhancing the energy consumption of water purification [78]. The fabrication of a graphene oxide–silver nanoparticles (GO–AgNPs) composite on a cellulose acetate membrane generated an antibiofouling membrane. It was observed that the GO–AgNPs composite on the membrane possessed a strong antibacterial activity to remove bacteria from water [79].

Toxic organic contaminants such as polycyclic aromatic hydrocarbons (PAHs), polychlorinated biphenyls (PCBs), and dichloro-diphenyl-trichloroethane (DDT) are found to degrade very slowly, be persistent, and bioaccumulate. These pollutants are generated from domestic sewage, urban run-off, and effluents from various industrial and agricultural actions, such as food processing, pesticides, pulp and papermaking, and farming. These contaminants are lipid-solvable, teratogenic, carcinogenic, and neurotoxic pollutants. Therefore, such contaminants must be eliminated from waters because of their toxicity and harmful effects to water uses for different purposes, such as drinking, household needs, recreation, and fishing [78]. PAH contaminants such as naphthalene, phenanthrene, and pyrene are organic compounds composed of carbon and hydrogen, containing three or more fused benzene rings with delocalized π electrons. The main cause of PAH contaminants is the petroleum and petrochemical industry through oil leakage and disposal [76]. GO displays a high affinity for PAHs. A study was conducted to investigate whether PAHs could be adsorbed onto GO from water efficiently [80]. Water samples were obtained from the lakes, seas, and rivers and treated with GO for 10 min to allow adsorption to occur. The amount of PAH impurities was compared before and after treatment. The results showed that GO adsorbed PAHs with high efficiency. After adsorption, the GO solution should be aggregated for the extraction of PAHs from water. Due to carboxyl group edges, GO sheets are highly negatively charged. Electrostatic repulsion stabilizes aqueous colloids and dispersions. GO dispersion and aggregation are determined by the ionization levels of its carboxylic acid and phenolic hydroxyl groups. To achieve GO aggregation, NaCl is used to neutralize excessive negative charges and lower the electrostatic repulsion. The addition of NaCl accomplishes GO aggregation and the extraction of PAHs. Many factors have been reported to enhance the adsorption of PAHs by GO. It was found that the mass of GO greatly affected its adsorption capacity for PAHs, which had an impact on the efficiency of the extraction. The adsorption measurements of GO for PAHs showed that, as the GO’s mass increased, the adsorption capacity increased. The same occurred with the efficiency of PAH extraction. Moreover, as the contact time increased in the range of 5–15 min, the PAH recovery also increased. As mentioned before, NaCl was used to neutralize the excess of negative charges and to lower the electrostatic repulsion that facilitated GO aggregation. For that reason, the NaCl concentration impact on the extraction efficiency was investigated. The results indicated that a lower concentration of NaCl caused GO aggregation and low extraction efficiency of the PAHs from water. However, a higher concentration of NaCl was favorable for low PAH recovery by GO due to the competition between Na+ and the PAHs, so the PAH recovery was decreased at both higher and lower concentrations of NaCl. A study of GO aggregation efficiency at various contact periods of time revealed that, as the disposition time increased, the recovery of PAH compounds increased. It is known that GO has high efficiency for PAH adsorption, but there are some chemicals, such as inorganic and organic interfacial materials, that may affect the adsorption efficiency of PAHs. In the seawater sample, the GO aggregation occurred directly when GO was added due to the high amount of salt, which resulted in poor PAH extraction. This was because the seawater must be diluted, and no NaCl addition is required to facilitate GO aggregation in this method. Extraction and separation through the aggregation of GO using NaCl as the electrolyte solution was reported to be environmentally friendly, simple, and inexpensive [80]. Polychlorinated dibenzofurans and biphenyls are greatly toxic and stable contaminants [70]. Fe_3_O_4_-nanoparticle-grafted graphene oxide (Fe_3_O_4_@GO) showed the ability for the removal of 2,4,4′-trichlorobiphenyl (PCB 28) pollutants from a large volume of water. It needed only 30 min to remove trace levels of PCB 28 contaminates from a 200 mL water sample using a magnetic solid-phase extraction (MSPE) technique based on Fe_3_O_4_@GO sorbents. Thus, it could be applied for cleaning polychlorinated biphenyl pollutants from water [81]. In addition, pristine graphene nanosheets and GO were used for the removal of biphenyl and phenanthrene in distilled and deionized water. For naphthalene, 2-naphthol, 1,2,4-trichlorobenzene, and 2,4,6-trichlorophenol removal from water, graphene and GO were used to achieve the greatest adsorption due to π–π interactions and hydrogen bonding between the hydroxyl groups of pollutants and oxygen-containing functional groups of the sorbents [78]. Fe/GO and Cu–Fe/GO nanocomposites were produced with an atomic implantation method for the photocatalytic degradation of DDT in water. It was observed that oxides of Cu and Fe were distributed equally on GO and were present as Cu^+^ and Fe^2+^ ions in the Cu–Fe/GO nanocomposite. Compared to Fe/GO, Cu–Fe/GO nanocomposites exhibited high photocatalytic DDT removal, reaching 99.7% at 0.2 g/L catalysts, 15 mg/L H_2_O_2_, and pH 5 through the generation of −OH radicals, which improved the DDT elimination ability [82].

### 5.3. Water Desalination Membranes

There is a shortage of fresh water in some regions of the world. The desalination of seawater is one of the solutions to this problem. Layered GO membranes (GOMs) have a controlled, subnanometer-wide interlayer distance and versatile surface chemistry, which gives GOMs the ability to accurately sieve small ions and molecules. Pristine and chemically improved GOMs efficiently blocked organic dyes and nanoparticles but were unsuccessful in blocking smaller ions with hydrated diameters [83]. To overcome this, it is possible to reduce the interlayer spacing down to only several angstroms to block small inorganic salt ions. However, compressed GOMs greatly decrease the water flux, limiting practical applications. Planar heterogeneous graphene oxide membranes (PHGOMs) are the latest approach to have been tested as an efficient water desalination process. The results have shown that PHGOMs possessed excellent salt elimination ability and high water flux. PHGOMs are composed of pristine, negatively charged GOs (n-GOs) and polyethyleneimine-conjugated, positively charged GOs (p-GOs). Horizontal ion transport through oppositely charged GO multilayer lateral hetero-junctions exhibits a bi-unipolar transport manner, which stops the conduction of both cations and anions. The salt concentration is depleted in the near-neutral transition area of the PHGOM with the help of a forward electric field, and deionized water can be extracted from the depletion zone. This approach gave a great NaCl rejection rate, reaching 97.0%, and very high water flux through an inverted T-shaped water extraction mode [83]. It was reported that the GO membrane alone was permeable to water but impermeable to other molecules and impurities, such as bacteria, gases, vapors, and metal ions, but had poor salt elimination and water flux. In addition, another obstacle was observed: GO membranes swelled when immersed in water. An rGO membrane with the same laminated structure and high stability in water was found to resolve this obstacle by decreasing the rGO membrane thickness, which in turn increased the permeability of the rGO to impurities. Liu et al. observed that freestanding, ultrathin rGO membranes (thickness range of 200–20 nm) treated with hydriodic acid exhibited high salt rejection with fast water flux compared to GO membranes [84], but the latest and most effective water desalination approach consists of using a PHGO membrane [83].

### 5.4. Removal of Oil Pollution

Crude oil pollution (oil spill) is a serious worldwide problem that causes disastrous effects on the environment and ecosystem. Standard treatment methods such as in situ burning, manual skimming, and bioremediation require a great workforce and a long duration. In addition, crude oil possesses a high viscosity, which forms another obstacle for standard adsorbents [85]. One solution is a floating adsorbent composed of rGO produced by a facile, one-pot hydrothermal method, a melamine sponge (MS), and a 3D-printed mounting platform. rGO-MS composites have appealing hydrophobicityand oleophilicity for oil absorption in water at a contact angle of 122°. rGO-MS composites can absorb nearly 95 times their weight in crude oil in 12 min under light irradiation because of efficient light-to-heat conversion. However, a 3D-printed mounting platform for rGO-MS composites has been fabricated to enhance their performance for enhanced extraction. rGO-MS composites were successful for in situ crude oil removal due to rGO’s hydrophobicity, oleophilicity, and photothermal characteristics, as well as the MS’s floating capability [85].

The green, absorbent material created by the inclusion of rGO in natural rubber (NR) latex to synthesize an (NR/rGO) composite improved the petroleum oil (gasoline and crude AXL oil) adsorption capacity of the composite compared to pure NR foam and a cost-effective adsorbent. The NR/rGO composite had high elasticity and enhanced oil adsorption capacity. Additionally, the reusability of the adsorbent material for oil removal ability was greater than 70% after 30 uses. Moreover, environmental conditions such as temperature and ocean waves could affect the oil adsorption capacity of the NR/rGO adsorbent composite. It was observed that, as the temperature increased up to 45 °C or as an external force such as waves increased, the oil adsorption capacity of the adsorbent composite increased. It was concluded that the (NR/rGO) composite was a reassuring substitute for oil adsorbent in oil spill purification under serious field conditions in the ocean [86].

### 5.5. Cancer Treatment

In a recent study, aminated GO (GO-NH_2_) could activate powerful cytotoxic and genotoxic effects in colorectal cancer cells when compared to pristine (pure) GOs. The cytotoxicity of GO-NH_2_ in colon cancer cells consisted of various mechanisms, such as the induction of ROS production, the blockage of cell proliferation, increased cytotoxicity, the induction of DNA damage, and the initiation of apoptosis. These parameters were more efficient at the highest concentrations of GO, which resulted in higher ROS production. It was concluded that GO-NH_2_ nanoparticles were promising for the treatment of colon cancer [87]. In another study, GO was used as a multifunctional platform for therapeutic delivery, biological imaging, and cancer sensing due to its pH-influenced fluorescence emission in the visible and near-infrared spectra, which yielded possibilities for molecular imaging and pH sensing [88]. Additionally, GO is water-soluble and can be a platform for functionalization, allowing its use for drug delivery. Furthermore, GO showed a great cellular internalization capacity with lower depuration after 24 h, making it a good delivery agent. Being a pH-sensitive, fluorescent nanomaterial, GO can be utilized for the identification of the pH levels of the cellular environments. Thus, it can be potentially used for sensing the acidic extracellular environments of cancer cells [88]. It was reported that GO confined the growth of tumors in different cell lines, including ovarian, pancreatic, breast, and lung cancers, as well as glioblastoma. GO could promote toll-like receptor response and cause autophagy and antitumor effects [4].

Phenolic compound resveratrol was used for the green process of reducing GO to rGO, and its antitumor potential was tested against ovarian cancer cells. Dose-dependent effects were observed, including membrane leakage and oxidative stress. rGO was significantly more cytotoxic compared to GO, which could induce cell death in less than 60% of the A2780 cells even at the highest tested concentration, whereas rGO already caused significant cytotoxicity at 20 μg/mL; at 80 μg/mL, 90% of the cells were dead [89].

Graphene nanocomposites have also been developed for this purpose. Composite rGO–AgNPs showed stronger antitumoral effects when compared to other tested nanomaterials such as graphene oxide, rGO, and AgNPs. This composite inhibited cell viability in A2780 ovarian cancer cells and increased lactate dehydrogenase leakage, reactive oxygen species generation, caspase-3 activity, and DNA fragmentation [90]. The inhibition of the cell viability was dose-dependent. While the IC_50_ for GO, rGO, and AgNPs were ~60 μg/mL, ~25 μg/mL, and ~20 μg/mL, respectively, for the nanocomposite (rGO–AgNPs), the IC_50_ was only ~12.5 μg/mL, showing that the association of this nanomaterial could increase antitumor activity [90]. In another study, similar nanocomposite rGO–AgNPs were tested against human lung cancer A549 cells, and the IC_50_ was only 30 μg/mL [91]. A GO-CuO nanocomposite demonstrated activity against HCT-116 human colon cancer lineage, leading to a 70% reduction in cell viability at the concentration of 100 μg/mL [92].

Additionally, graphene-based materials can be used as drug carriers or antitumor agents for photothermal therapy. Photodynamic therapy is a noninvasive treatment methodology to treat diseases such as cancer. In this process, the photosensitizer molecule transfers the photon energy to surrounding oxygen molecules for ROS production and is heated under the irradiation of light with appropriate wavelengths [93,94]. This approach allows a selective effect since only the lesion exposed to the light and to graphene-based nanomaterials is treated. GO-PEG-Ce6 showed excellent water solubility and caused cytotoxic effects on human nasopharyngeal epidermal carcinoma KB cells under light excitation. This complex enhanced the antitumoral potential when compared to free Ce6 photosensitizers due to the cell uptake of Ce6 delivered by graphene [93]. rGO with a noncovalent PEG coating could eliminate 4T1 tumors in mice after an intravenous injection of rGO-PEG with a dose of 20 mg/kg and under 808 nm laser irradiation (0.15 W/cm^2^). However, under the same conditions, GO-PEG could not inhibit tumor growth. All the mice treated with rGO-PEG survived over 100 days without a single death, side effect, or tumor regrowth [94]. In addition, the GO-Fe_3_O_4_ composite was modified with PEG and cetuximab, an antibody for the epidermal growth factor receptor (EGFR). The composite could carry the anticancer drug doxorubicin (DOX) for a pH-dependent release. Additionally, the composite could inhibit the CT-26 murine colorectal cell growth in both in vitro and in vivo assays in photothermal therapy [95]. GO modified with hyaluronic acid (HA) was developed for photothermal therapy against cancer cells. HA specifically binds to the CD44 receptor, which is abundantly overexpressed on the surface of various cancer cells, allowing more specific activity against tumor cells. After laser irradiation treatment (808 nm), GO-HA at a concentration of 50 μg/mL for 24 h was able to inhibit 48.04% of human breast cancer MCF-7 cell lines [96].

### 5.6. Bone and Teeth Implantation

Titanium (Ti) surface improvement was achieved using GO coating and aspirin (A) loading (A/Ti-GO). Ti-GO was synthesized using an alkali-hydrothermal reaction and a coupling agent. The torsion test revealed that there was stable bonding between the GO coating and Ti under a torsional shear force in clinical settings, and there was no falling off of the GO coating from the sample surface (good adherence). In addition, the release of aspirin loaded on the Ti-GO surface was retained for 3 days due to π–π stacking interactions. In vitro cell studies showed that A/Ti-GO facilitated the proliferation of MC3T3-E1 cells and osteogenic distinction. The A/Ti-GO surface can be beneficial in enhancing the success rate of Ti implants in patients with bone conditions such as diabetes and osteoporosis [97]. In recent studies, GO, lysozyme (Lys), and tannic acid (TA) have been combined using an easy and controlled layer-by-layer technique to produce a powerful antibacterial and modified osteogenic multilayer coating. Coatings with antibacterial and osteogenic agents can be useful in dental implants. GO, Lys, and TA coatings possess physical characteristics such as wettability, roughness, stiffness, and continual growth with the deposited process. Additionally, the obtained coatings showed improved osteogenesis of dental pulp stem cells (hDPSCs), which determined the potential application of coatings for dental implants [98].

### 5.7. Scaffolds for Mammalian Cell Culture

Due to GO properties that advance cell adhesion and growth, graphene-based material substrates are a good prospect for tissue engineering and cell scaffolds. Scaffolds are structures able to support living cells that create a suitable microenvironment, which enables cells to grow and maintain themselves. Scaffolds need a high level of porosity, allowing the passage of nutrients and oxygen into the system for maintaining the cells, as well as the possibility to release the waste products and therapeutic products secreted by the cells [99].

Various oxidized graphene-based papers can be used as substrates for cell culture by using starting materials with various thicknesses and lateral dimensions. Graphite oxide has thicker sheets compared to the thinner, large graphene oxide (l-GO) and small graphene oxide (s-GO) sheets. These substrates were tested for their cellular adhesion and proliferation ability with two epithelial cell lines, human lung cell culture (A549) and human neural cell culture (SH-SY5Y); the variances in their morphologies were observed using microscopic analysis. These GO sheets promoted cell growth with no impact on cell adhesion, proliferation, and morphology. In addition, although these three GO materials exhibited different topographies, they had similar structural and physicochemical characteristics. This concluded that paper-based GO substrates were successful biocompatible cellular materials that promoted anchorage-influenced cell growth and had the potential to be researched further for utilization in tissue engineering, regenerative medicine, substrates for cell growth, and bionic applications [100]. Another study indicated that GO film was effective for regulating the structure and function of human adipose-derived stem cells (hASCs). GO films fabricated through the utilization of a self-assembly method presented appropriate conditions for the adhesion, affinity, proliferation, and distinction of hASCs. Good attachment and higher affinity of the hASCS were observed on GO film compared to an uncoated substrate. Moreover, time-dependent cell viability of the hASCs occurred on the GO film. The GO films improved the differentiation of hASCs such as osteogenesis, adipogenesis, and epithelial genesis, but they reduced the chondrogenic differentiation of the hASCs. The GO films were a successful substrate for hASC cultures and could be used in designing and manipulating scaffolds for biological, stem cell, and tissue-engineering applications [101].

To use graphene, it is necessary to evaluate its cytotoxic potential, which can alter metabolic functions and proliferative capacity and can induce apoptosis. In addition, it is necessary to evaluate the possible transformations after the interaction with cells, such as aggregation and changes in lateral size. In addition, it is also important to assess the potential to induce inflammatory responses when administered in vivo [99].

### 5.8. Biofunctionalization with Proteins and DNA

GO has a larger surface area and is rich in oxygen content, which enhances the immobilization process and detection sensitivity. A GO-based sensor can detect many targets, such as single-stranded DNA (ssDNA), living cells, and metal ions [4]. Proteins can be immobilized onto GO without the need for surface modifications or coupling reagents and can be applied to the detection process [102]. The amphiphilic protein hydrophobin was successfully immobilized on GO sheets at the hydrophobic surface [103]. The attachment of amino Fe_3_O_4_ onto GO through covalent bonds produced magnetic GO, which was used in the immobilization of laccase, resulting in higher thermal stability and different pH values [104]. The immobilization of naringinase with graphene resulted in high isoquercitrin production. In addition, it was observed that, when graphene sheets were immobilized with an enzyme, it increased the specificity, and a moderate catalytic characteristic was observed so that the enzyme could be reused [105].

Green fluorescent protein (GFP) is biologically inert and releases bright green, fluorescent radiation upon exposure to blue ultraviolet light [4]. It is utilized in the identification of cells and tissues that have targeted gene expression [106]. The incubation of cells with GFP-rGO did not impact cell morphology, indicating that the composite was nontoxic for the cells and had the potential to be utilized as a biomarker to investigate the cytotoxicity and identification of cells or areas of tissues possessing an expression of target genes [107]. The ssDNA could be effectively adsorbed on the graphene surface due to the big, 2-D, aromatic surface of graphene. The ssDNA–graphene biointerface was utilized in a field-effect transistor for the label-free and reversible identification of complementary ssDNA. Additionally, ssDNA adsorbed onto GO was used for studying surface-enhanced laser desorption ionization time-of-flight mass spectrometry [102].

### 5.9. Biosensing and Bioimaging

Graphene-based materials, due to their high sensitivity, inexpensive, fast response and simple operation, are utilized in the fabrication of biosensors based on various sensing methods, such as optical and electrochemical signaling. These materials are successful electrode materials due to their electrochemical characteristics, which can enhance the detection of biomolecules [108] such as thrombin, oligonucleotides, ATP, amino corrosives, and dopamine [4]. Biomolecules have an essential duty in all life activities, such as disease development, so the precise identification of biomolecules is necessary for disease diagnosis and therapy. GO is incorporated in the fabrication of biosensors due to its excellent optical properties, such as its ability to fluoresce over a broad range of wavelengths (from near-infrared to ultraviolet) and efficiently quench the fluorescence of other fluorescent dyes. FRET is a well-developed technology for DNA identification and various atoms [108] and for measuring nanometer-scale distance and changes, both in vivo and in vitro [102]. GO is used as an energy acceptor in FRET biosensors, but it can be used as the energy donor or acceptor in various immunosensors [108].

In a recent study, an aptamer-carboxyfluorescein (FAM)/GO nanosheet complex could detect ATP and GTP in functional cells. The aptamer was shielded from enzymatic digestion, and the FAM fluorescence was extinguished through absorbing onto the GO surface. After uptake by JB6 cells or a human breast cancer MCF-7 cell, the identification between the aptamer and the intracellular ATP or GTP caused conformation changes in the aptamer structure, resulting in the restoration of FAM fluorescence [109]. Magnetic resonance imaging (MRI) is an important in vivo and noninvasive imaging technique that is employed in clinical practice. Aminodextran-coated Fe_3_O_4_ nanoparticles were immobilized onto GO to improve biocompatibility and cellular MRI signals. Additionally, it had excellent physiological stability, low cytotoxicity, and could be internalized by HeLa cells [110]. Nuclear medical imaging positron emission tomography is capable of quantifying radioisotope concentrations in vivo with great tissue penetration. GO united with a TRC105 antibody has been utilized as an in vivo tomography imaging agent for CD105 (a biomarker for tumor angiogenesis) targeting [108,111].

### 5.10. Gene Delivery and Drug Delivery

Gene therapy is an assuring approach for the treatment of different diseases caused by genetic disorders. Studies have determined that graphene and GO have great potential to be utilized as gene carriers because ssDNA and RNA can immobilize onto graphene and GO with noncovalent adsorption through stacking, electrostatic, and other molecular interactions. GO–PEI (polyethylenimine) is an ideal gene vector due to its lower cytotoxicity and advanced transfection efficiency at an optimal mass ratio. PEI–GO possesses the ability to condense DNA at a low mass ratio with a positive potential and to transport plasmid DNA into cells and be localized in the nucleus efficiently [112,113].

Graphene nanomaterials have an ultrahigh surface area and sp^2^-hybridized carbon area, making them effective drug carriers to load a high amount of drug molecules on both sides of a single atom-layer sheet. They have many chemically reactive oxygen-containing groups on their surfaces that can be used for the functionalization of diverse compounds via covalent bonding from carboxylic acid, epoxy, and hydroxyl groups. Beyond that, GO sheets also exhibit noncovalent binding with some molecules via hydrophobic interaction, π–π interaction, or van der Waals interaction on sp^2^ networks that are not oxidized [114]. Functionalization can improve different properties, such as solubility, stability, and specificity, as well as cellular uptake, by enhancing its ability to go across the target cell membrane [114]. In addition, it is possible to adopt different approaches for the controlled release of drugs, such as changes in pH, enzymatic action, reducing environment, or electrostatic interactions (Figure 5) [114,115,116].

The anticancer drugs SN38 and DOX can be loaded onto GO by simple physisorption through π–π stacking to target cancer cells. PEI is covalently bonded to GO through a simple amidation process. PEI-GO assists in loading siRNA (small interference RNA), which restricts protein expression by targeted the cleavage of mRNA through electrostatic adsorption and aromatic anticancer drugs (such as DOX and camptothecin) through π–π stacking, resulting in improved anticancer efficiency [108].

GO was functionalized with methotrexate, an anticancer drug, to assess its potential as a carrier for the delivery of anticancer drugs. GO did not cause any considerable cytotoxic effect to any of the cells tested: hepatocellular carcinoma cells (HepG2 cells), human embryonic kidney cells (HEK293A cells), and porcine skin fibroblasts (PEF). On the other hand, methotrexate was toxic to all the cells without any apparent selectivity. However, GO-methotrexate displayed significant, specific cytotoxicity to the tumor cell line (HepG2 cells) when compared to normal cells [117]. Furthermore, changes in the pH of the tumor environment can be used for a drug delivery approach. In this regard, GO was functionalized with carboxymethyl cellulose (CMC) and then DOX to form GO-CMC/DOX. The release rate of the drugs reached an optimum value of 65.2% at pH 5. The DOX and GO-CMC drug carriers were bound with π–π stacking action and a hydrogen bonding interaction. The π–π bond at the lower pH environment was broken, which led to the slow release of the drug [118].

## 6. Environmental Toxicity

GO-based materials can be used for many applications, causing an increment in the release of nonbiodegradable GO into the environment [119]. Biofilms have an important role in the ecosystem function and assist in organic matter decay and biogeochemical cycling [120]. GO accumulation on biofilms results in the reduction in bacterial activity and viability in soil. In addition, GO decreases the bacterial metabolic activity and viability, as well as restricts essential microbial functions that are required in activated sludge processes such as the elimination of organic matter and nutrients such as nitrogen and phosphorus [37].

The aquatic environment usually is the final destination of a variety of effluents’ disposal. Once in water bodies, graphene derivatives can interact with different inorganic ions and molecules, and hence, affect the ecosystem [121]. Graphene-based nanoparticles (GPNs) can accumulate in an aquatic environment, causing problems to aquatic habitats and food chains. GPNs are toxic to the cellular environment, even at very low concentrations, and their toxicity appears to be size-dependent. Smaller nanoparticles are more toxic compared to larger ones. A, GPN surface properties have an important influence on toxicity. A study conducted using embryos, larvae, and adult zebrafish, as well as embryos and larvae of Japanese medaka, indicated that the toxic effects of GPNs were mainly caused by oxidative stress because of ROS production. It was highlighted that further studies should be conducted to investigate GPN toxicity in aquatic environments [122]. The production of high-performance rGO can generate toxic substrate emission; in particular rGO produced with Hummer’s method emits harmful chemical substances to the environment. Some of these substrates, such as hydrazine, are toxic and affect the environment and animals’ health. The emission of NOx influences photochemical ozone formation, acidification, terrestrial eutrophication, and marine eutrophication. Chloride and nitrate are also known to be harmful compounds [123]. At low concentrations, GO did not affect the zebrafish embryonic development, but at a higher concentration, it induced significant embryonic mortality. Moreover, it caused increased heartbeat and apoptosis, delayed hatching, cardiotoxicity, and cardiovascular defects, and it decreased hemoglobin production [124]. Additionally, GO caused acute and chronic toxicity to the freshwater cladoceran *Ceriodaphnia dubia*. Chronic exposure decreased the number of neonates and the feeding rates and increased ROS generation [125].

In adult zebrafish, exposure to large, ^14^C-labeled few-layer graphene (FLG) enhanced the graphene amount by more than 170-fold when compared to smaller FLG at the same concentration. In addition, smaller FLG was able to pass through the intestinal wall and enter the intestinal epithelial cells and blood. Large FLG was excreted to more than 95% after 4 h of depuration. And after 120 h, the graphene content was not significant. In contrast, zebrafish exposed to smaller FLG could eliminate 30% of the graphene amount after 4 h. The graphene content remained unaltered after 68 h and accumulated in the gut [126]. GO induced higher ROS production and accumulated in the gut of *Daphnia magna*. However, almost all the GO was eliminated in 24 h. The easier depuration could be related to GO’s higher dispersibility and hydrophilicity [127]. In addition, ^14^C-labeled FLG accumulated in mice livers, where larger FLG induced damage to red blood cells and, consequently, caused phagocytosis with the Kupffer cells. In these cells, the hemoglobin degradation released iron that enhanced the Fenton reaction, producing hydroxyl radicals. Under these conditions, the larger graphene was degraded into CO^2^ [128]. Similarly, rice plants (*Oryza sativa*) accumulated FLG in their roots and shoots. After 14 days, more than 70% remained in the plant, and 9% of the FLG was degraded to CO^2^ in a Fenton reaction [129].

## 7. Cytotoxicity

GPNs can penetrate through cellular barriers, enter cells, and interact with almost all of the cellular sites, such as the plasma membrane, cytoplasmic organelles, and nucleus. GPN interaction with DNA may harm the genome and epigenome [122]. Recent studies have determined that graphene-based materials can be toxic to organisms such as bacteria, nematodes, zebrafish, and humans. Cytotoxicity toward bacteria via both membrane and oxidative stress has been observed for both GO and rGO, and oxidation levels affect cytotoxicity. The results of a recent study showed that GO nanosheets promoted toxicity against Gram-negative and Gram-positive bacteria through the production of ROS [12]. The ROS generation of GPNs could be mainly responsible for processes such as apoptosis, metabolic disorders, neurodegeneration, and immunomodulation [122]. A recent study investigated GO’s size effects on Leydig (TM3) and Sertoli (TM4) cells by considering two different nanosheets (100 and 20 nm) prepared with a modified Hummer’s method. The 20 nm GO promoted a great loss of cell viability and cell proliferation, extensive leakage of lactate dehydrogenase (LDH), and ROS generation compared to the 100 nm GO. However, both sizes of GO promoted a reduction in the mitochondrial membrane potential in TM3 and TM4 cells. Both GOs generated oxidative destruction to the DNA by elevating the 8-oxo-dG levels, which are produced in DNA damaged by ROS. Additionally, they upregulate different genes that control DNA damage and apoptosis. The results revealed that the 20 nm GO had more toxicity compared to the 100 nm GO, and the reductions in MMP and apoptosis were the primary toxicity behaviors of both GOs. GOs possess size-dependent germ cell toxicity in male somatic cells, especially TM3 cells [130]. The cytotoxic impact on MCF-7 cells was evaluated at various dosages of GO-ZnO. The results showed that GO-ZnO exhibited cell-killing behavior at higher concentrations of GO with a loss in cell viability in a dose-dependent manner. Additionally, GO-ZnO promoted oxidative stress and cytotoxicity due to ROS production [131]. Srikanth et al. determined that GO promoted cytotoxicity and oxidative stress in blue sunfish cells (BF-2). In addition, it was observed that GO promoted dose- and time-dependent cytotoxicity in BF-2 cells. Moreover, the exposure of BF-2 cells to higher concentrations of GO for longer hours increased oxidative stress and induced greater cytotoxicity due to the higher generation of ROS [132].

The skin is one of the organs most exposed to graphene-based materials, so it is crucial to investigate the effects of graphene-based materials on skin cells. The influences of four graphene-based materials, an FLG produced with a ball-milling treatment and three GO compounds (GO1, GO2, and GO3), were evaluated on HaCaT keratinocytes in an in vitro model of skin toxicity. The results showed that, after a 72 h exposure duration to FLG, the less-oxidized compound had lower cytotoxic effects, resulting in harm to the mitochondrial and plasma membranes. On the other hand, the GO3 compound was the highest-oxidized and the highest in cytotoxicity, promoting greater harm to the mitochondrial and plasma membranes. It was concluded that larger concentrations of and long exposure durations to FLG and GOs could reduce mitochondrial function and damage the plasma membrane, indicating a cytotoxic impact at the skin level [133]. In another study, the mitochondrial activity in adherent human skin fibroblasts was assessed after exposure to graphene sheets and GO. The results showed that compacted graphene sheets were more harmful to mammalian fibroblasts when compared to low, densely packed graphene oxide [134].

Another study reported that the cytotoxicity of graphene and GO on red blood cells could be evaluated by measuring the efflux of hemoglobin from these cells. Higher hemolytic activity was observed at the smallest size of GO particle, and lower hemolytic activity was observed by aggregated graphene sheets. The particle size, particulate state, and oxygen concentration or surface charge of graphene had a great influence on biological or toxicological reactions to red blood cells [134]. Wang et al. investigated the impact of GO and nitrogen-doped graphene quantum dots (N-GQDs) on the same cells by assessing their hemolytic activity, examining the morphological changes and observing the ATP content after exposure to GO and N-GQDs nanomaterials. The adsorption of GO caused damage to the membrane integrity by removing the lipid bilayer, which caused hemolysis and aberrant forms of the cells. However, N-GQDs only disrupted the structure and conformation of the lipid, leading to an aberrant form of the cells [135]. Yang et al. demonstrated that monolayer and multilayer GO facilitated ROS generation in dendritic cells. However, monolayer GO had a lower impact on cell viability than multilayer GO. Additionally, both types of GOs induced immunotoxicity and cell disruption. Gene expression profiling determined that both GOs generated changes in the transcriptome and that monolayer GO generated more altered genes compared to multilayer GO [136].

## 8. Conclusions

GO exhibits numerous fascinating and unique characteristics, such as hydrophilicity, high dispersion in aqueous media, easy synthesis, robust size, high biocompatibility, and surface functionalization ability, due to the presence of functional groups. These allow promising applications in various fields, including biological and biomedical areas. Synthesis methods have been modified throughout the last two decades to reduce the toxicity of the process. Moreover, the modifications have allowed simple and less expensive processes. Conventional methods release harmful by-products and toxic reagents, which limit GO and rGO applications. Moreover, GO’s characteristics strongly depend on the synthesis method, especially the size and surface charge, which can be controlled by changing the oxidation temperature through a modified Hummer’s method. Although it has many benefits, the toxicity of GO needs to be evaluated before any use. Several studies have reported negative effects of GO and rGO on biological systems, including human health. Regarding the environment, graphene-based materials can accumulate and affect all living organisms. The cytotoxic effects have been found to be dependent on intrinsic factors such as size, dosage, exposure time, and the functionalized compounds. However, by manipulating these factors, it is possible to apply graphene-based nanomaterials under safe conditions. Furthermore, in-depth and broad understandings of the environmental transformations, bioaccumulation, chronic exposure effects, and depuration in living organisms are highly needed.

## Figures and Tables

**Figure 1 jfb-13-00077-f001:**
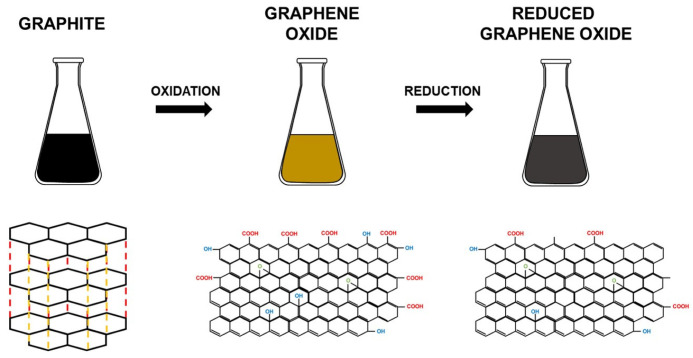
Representation of the GO and rGO production process and chemical structure.

**Figure 2 jfb-13-00077-f002:**
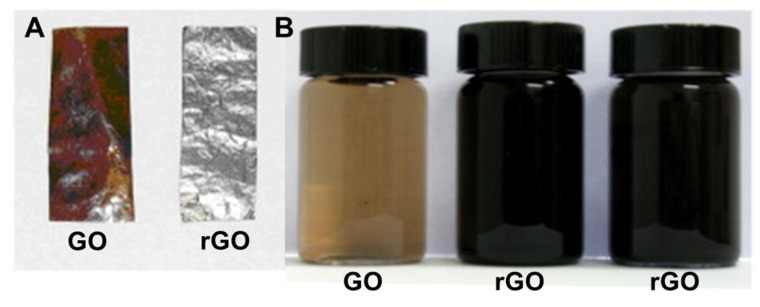
(**A**) Typical coloring of GO and rGO films [19] (Copyright © 2022 Elsevier). (**B**) Typical colorings of GO (yellow-brown) and rGO (black) solutions [20] (Copyright © 2022 Elsevier B. V.).

**Figure 3 jfb-13-00077-f003:**
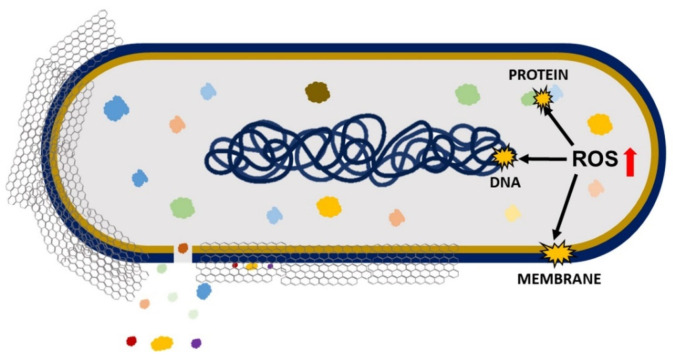
Antimicrobial mechanisms of GO and rGO involving oxidative and membrane stress that result in DNA, protein, and membrane damage.

**Figure 4 jfb-13-00077-f004:**
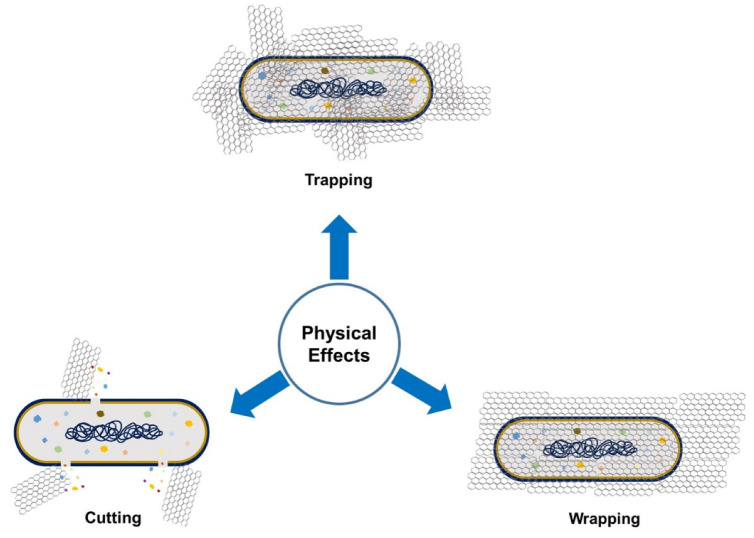
Physical factors that, through graphene-derivative microorganism interaction, can affect microbial growth.

**Figure 5 jfb-13-00077-f005:**
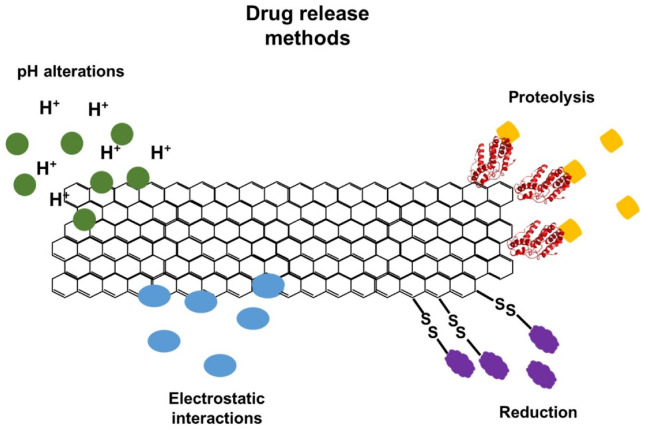
Different methods that can be applied for the controlled release of drugs functionalized in graphene and derivative sheets.

**Table 1 jfb-13-00077-t001:** Antimicrobial activity of graphene derivatives on different microorganisms.

Microorganisms	Type	Nanomaterial	Evaluation Method	Dose	Antimicrobial Activity	Ref.
*Bacillus subtilis*	Gram-positive bacteria	rGO	Optical density	1 × 1 cm^2^ graphene-based membrane	No growth detected	[38]
		rGO	Agar well diffusion	0.1–0.8 mg/mL	1–3.5 mm	[39]
		GO	Microdilution	25–200 μg/μL	48.86–91.40%	[40]
*Escherichia coli*	Gram-negative bacteria	GO	Colony counting	40 μg/mL	69.3 ± 6.1%	[41]
		rGO	Colony counting	40 μg/mL	45.9 ± 4.8%	[41]
		GO	Colony counting	25–150 μg/mL	18–87%	[20]
		rGO	Colony counting	25–150 μg/mL	14–81	[20]
		GO	Colony counting	3 mg/mL	80%	[42]
		rGO	Agar well diffusion	0.1–0.8 mg/mL	2–5 mm	[39]
		GO	Agar well diffusion	1 μg/μL	39 mm	[43]
		GO	Optical density	62.5–500 μg/mL	~40–95%	[44]
*Fusarium graminearum*	Fungal	GO	Germination spores	10–500 μg/mL	21.66–85.48%	[45]
*Fusarium oxysporum*	Fungal	GO	Germination spores	10–500 μg/mL	17.31–81.16%	[45]
*Klebsiella pneumoniae*	Gram-negative bacteria	GO	Agar well diffusion	1 μg/μL	41 mm	[43]
		GO	Microdilution	25–200 μg/μL	50.76–92.80%	[40]
		GO	Optical density	62.5–500 μg/mL	71.8–96.8%	[44]
*Proteus mirabilis*	Gram-negative bacteria	GO	Agar well diffusion	1 μg/μL	27 mm	[43]
*Pseudomonas aeruginosa*	Gram-negative bacteria	GO	Optical density	25–200 µg/mL	0–100%	[46]
		rGO	Optical density	25–200 µg/mL	0–100%	[46]
		GO	Growth curve	1–5 mg/mL	Up to 78.7%	[47]
		rGO	Growth curve	1–3 mg/mL	90.3–93.3%	[47]
		GO	Agar well diffusion	1 μg/μL	38 mm	[43]
		rGO	Optical density	1 × 1 cm^2^ graphene-based membrane	No growth detected	[38]
		GO	Optical density	62.5–500 μg/mL	~30–Above 95%	[44]
*Pseudomonas syringae*	Gram-negative bacteria	GO	Optical density	10–500 μg/mL	5–88.8%	[45]
*Salmonella typhi*	Gram-negative bacteria	GO	Microdilution	25–200 μg/μL	44.28–90.71%	[40]
*Serratia marcescens*	Gram-negative bacteria	GO	Agar well diffusion	1 μg/μL	39 mm	[43]
*Staphylococcus aureus*	Gram-positive bacteria	GO	Growth curve	1–3 mg/mL	Up to 93.7%	[47]
		rGO	Growth curve	1–3 mg/mL	Up to 48.6%	[47]
		GO	Agar well diffusion	1 μg/μL	38 mm	[43]
		GO	Microdilution	25–200 μg/μL	51.36–92.12%	[40]
*Streptococcus mutans*	Gram-positive bacteria	GO	Colony counting	12.5–50 μg/mL	Up to 80%	[48]
*Xanthomonas campestris pv. Undulosa*	Gram-negative bacteria	GO	Optical density	10–500 μg/mL	6.96–86.8%	[45]

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
