# Peer review of "Graphene Oxide (GO) Materials—Applications and Toxicity on Living Organisms and Environment"

_jfb, 2022, doi:10.3390/jfb13020077_

Round 1

Reviewer 1 Report

The manuscript Graphene Oxide (GO) Materials - Applications and Toxicity on 2 living Organisms and Environment.I suggest accepting this manuscript after the revision, and the authors should consider the suggestions described below:
The paper needs to be improved in the following manners, this is a nice study however, the following questions are necessary to be answered before further processing

1) Authors should carefully revised and corrected all the grammatical issues and follow the scientific norms in the whole manuscript

1) Resolutions of Figures The resolution of Figures and recheck.

3) Please use updated and recent papers in the literature review to give more sense to the reader.

4) Conclusions could be more specific and to the point, I would suggest looking and thinking about them.

5) Please more elaborate on the novel aspect of your work at the end of the introduction.

6) A details abbreviations section should be updated.

7) Some of the introductions on antibacterial and cells applications and … are need using from the below papers, so, below papers are add to the manuscript

Recent publications

1: Green synthesis of silver nanoparticles toward bio and medical applications: review study

2: Applications of graphene oxide in case of nanomedicines and nanocarriers for biomolecules: review study

Author Response

The manuscript Graphene Oxide (GO) Materials - Applications and Toxicity on 2 living Organisms and Environment. I suggest accepting this manuscript after the revision, and the authors should consider the suggestions described below:

The paper needs to be improved in the following manners, this is a nice study however, the following questions are necessary to be answered before further processing

1) Authors should carefully revised and corrected all the grammatical issues and follow the scientific norms in the whole manuscript 

Reply: We thank the reviewer for the comment. We revised all the manuscript’s grammar and content to reach the publications’ standard.

2) Resolutions of Figures The resolution of Figures and recheck. 

Reply: We thank the reviewer for the comment. All figures were revised and new versions with high resolution were provided.

3) Please use updated and recent papers in the literature review to give more sense to the reader. 

Reply: We thank the reviewer for the comment. As per reviewer recommendation, new up-to-date references were added.

4) Conclusions could be more specific and to the point, I would suggest looking and thinking about them. 

Reply: We thank the reviewer for the comment. The conclusion was revised and rewritten.

5) Please more elaborate on the novel aspect of your work at the end of the introduction .

Reply: We thank the reviewer for the comment. We revised the introduction and abstract and added more information in order to clarify the novelty of the present research review.

6) A details abbreviations section should be updated. 

Reply: We thank the reviewer for this suggetion. We added at the end of the munuscript an abbreviations section.

7) Some of the introductions on antibacterial and cells applications and … are need using from the below papers, so, below papers are add to the manuscript 

Recent publications “Green synthesis of silver nanoparticles toward bio and medical applications: review study”, “Applications of graphene oxide in case of nanomedicines and nanocarriers for biomolecules: review study”.

Reply: We thank the reviewer for the suggestions. We added both articles in the revised manuscript.

Reviewer 2 Report

In this study, the authors reviewed recent advances in carbon-based 2D materials and their applicability for environmental and biomedical applications. In general. the article can be reconsidered for publication after a major revision. 

1) The manuscript can be revised one more time, also some sentences are repetitive. 

2) So called "cutting effect" can be also observed in case of biomedical applications - thus, carbon-based 2D materials can be a dangerous! 

3) Carbon-based 2D materials are non-digestible, how these relatively big structures  can be eliminated from body? Again - how to minimize physical damage from them (cutting effect)? 

4) In general, black 2D materials can be used for photothermal therapy as well - this part is missed! 

Author Response

1)        The manuscript can be revised one more time, also some sentences are repetitive. 

Reply: We thank the reviewer for the comment. We revised all the manuscript’s grammar and content, to achieve the publications’ standard.

2) So called "cutting effect" can be also observed in case of biomedical applications - thus, carbon-based 2D materials can be a dangerous!  

Reply: This information is very appropriate, and the authors are very thankful for this comment. This is because numerous works cite that graphene and its derivatives cause toxicity in animals and cells. However, this danger is directly related to the given dose. Indeed, several in-vivo studies demonstrated that the GO cutting effect did not exhibit toxicity at a low dose (0.1 mg) and medium-dose (0.25 mg) but induced chronic toxicity at a higher dose (0.4 mg) as observed in the article by Wang et al.

(https://link.springer.com/article/10.1007/s11671-010-9751-6).

Some studies also reinforced this statement, i.e., increasing GO concentrations are dangerous and that the adjustment of exposure time and a low dosage may be favorable for safe and promising use of GO without any associated dangers being more significant than the benefits.

3) Carbon-based 2D materials are non-digestible, how these relatively big structures can be eliminated from body? Again - how to minimize physical damage from them (cutting effect)?  

Reply: We thank the reviewer for the comment. Yes, we entirely agree with this insightful information. However, it is possible to minimize the adverse effects of physical damage due to the cutting effect through the low dose and exposure time relationship. Moreover, Lu et al. (2017), reported that bigger structures are depurated from the organism. For this reason, the depuration time is important to understand the toxicity. Lv X. et al. (2018), demonstrated that GO was almost completely eliminated from the gut of Daphnia magna. The easier depuration could be related to the higher GO's dispersibility and hydrophilicity. Besides, Lu et al. (2021) and Huang et al. (2018), demontrated that part of graphene content could be degraded into CO2 by Fenton reaction. To clarify this on the manuscript, we added more information in enviroment toxicity section.

4) In general, black 2D materials can be used for photothermal therapy as well - this part is missed! 

Reply: We thank the reviewer for the comment. We cited the photothermal therapy for anticancer applications in Cancer treatment session. However, we added more information about this approach in this section.

Round 2

Reviewer 2 Report

A revised manuscript can be accepted for publication